# Climate Change and the Caribbean: Challenges and Vulnerabilities in Building Resilience to Tropical Cyclones

**Clint T. Lewis**

Department of Natural Resources and Environmental Studies, National Dong Hwa University, Hualien 974, Taiwan; clintlewis784@gmail.com

**Abstract:** Caribbean Small Island Developing States (SIDS) is one of the most vulnerable regions in the world to the impacts of climate change. The region has prioritized adaptation to climate change and has implemented many adaptation actions over the past 20 years. However, the region is becoming increasingly vulnerable to the impacts of tropical cyclones (TC). This paper analyses the impacts of TC on the region between 1980 to 2019. It aims to examine the economic loss and damage sustained by the region, identify the sectors most impacted, and ascertain the perspectives of key stakeholders on the factors that hinder building resilience. Statistical analysis techniques and semi-structured interviews were to unpack and understand the dataset. The paper finds that economic loss and damage has gradually increasing between 1980 to 2009 with a drastic increase between 2010 to 2019. The paper highlights the agriculture, housing, transport, and utility sectors as the most impacted. The findings also call to attention the need for increased access to adaptation financing for SIDS, the disadvantages of the income status that hinders building resilience, and the need for increased Early Warning Systems. The paper recommends revising the per capita national income as an eligibility criterion for accessing concessional development finance assistance, a comprehensive EWS for the countries in the region, and consideration of debt relief for countries affected by TC.

**Keywords:** Caribbean; climate change adaptation; small island developing states; tropical cyclone; vulnerability; resilience

## 1. Introduction

Climate change is a daunting challenge to Small Island Developing States (SIDS). SIDS are categorized as the most vulnerable countries in the world to the impacts of climate change [1]. These distinct group of developing countries are already experiencing climate-related impacts such as sea-level rise, more intense tropical and extra-tropical cyclones, increasing air and sea surface temperatures, and changing rainfall patterns [1]. These impacts place additional strain on people's livelihoods, and threaten the cultural survival and wellbeing of island communities [2]. According to [3], with the constant warming of the planet, should the 1.5 °C threshold be exceeded, SIDS will suffer significant social displacement and economic and environmental losses.

Vulnerability is defined as the propensity or predisposition to be adversely affected [4]. SIDS special characteristics which include their physical size, proneness to natural disasters, the extreme openness of their economies and low adaptive capacity have classified them as highly vulnerable to multiple stressors, both climate and non-climate [1,5]. For example, rapid-onset events such as TC which are classified as tropical depression, tropical storm, and hurricanes cause an estimated $835 million-worth of damage annually in the Caribbean [6]. In 2019, hurricane Dorian, another example of recurrent extreme climate events devastated the Bahamas. Two years prior, it was hurricanes Irma and Maria wreaking havoc in the region. The national governments and other key stakeholders in Caribbean SIDS have prioritized climate change adaptation to reduce its vulnerabilities to the impacts of climate change. Caribbean SIDS have notably been actively engaged in observing and assessing climate variables and understanding a range of other adaptation actions [7].

Climate change adaptation is a "process of adjustment to actual or expected climate and its effects" [4]. Adaptation is costly [8] and therefore, poses a "significant financial and resource challenge" to SIDS [1], which are already faced with continuous and interrelated economic, environmental, social, and political constraints. As a result of such constraints, SIDS seek international financing to help their endeavors in reducing their vulnerability to climate change and strengthen their adaptive capacities and resilience [9]. The Caribbean has actively been implementing various adaptation and mitigation projects and programs at a national and regional level for the past 20 years through international aid that includes Official Development Assistance (ODA), donor funds, bilateral funds, and multilateral funds in response to climate change. The Caribbean Community Climate Change Center (CCCCC) and the Organization of the Eastern Caribbean States (OECS) are two of the main institutions that coordinate the Caribbean regional and sub-regional response to climate change. Figure 1 highlights some of the important achievements towards building resilience to climate change through adaptation and mitigation actions.

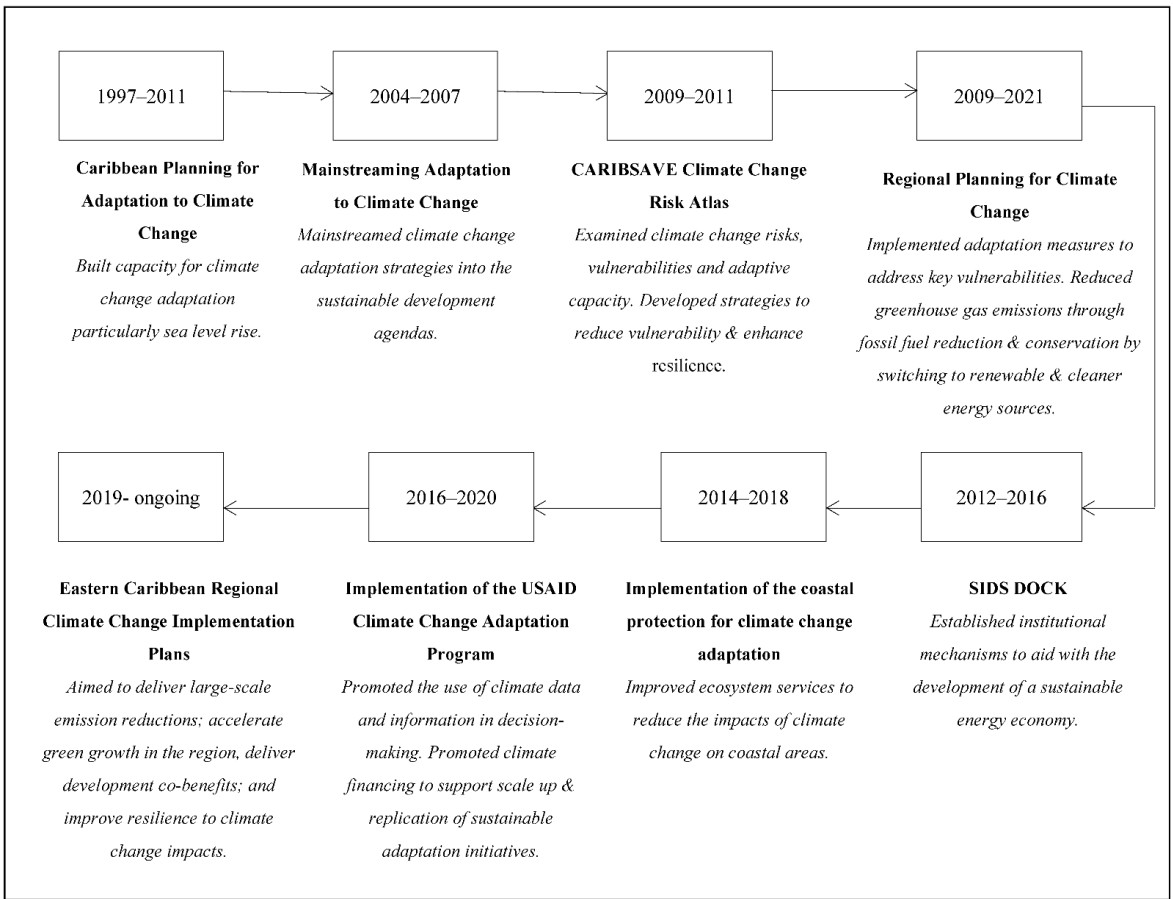

**Figure 1.** Eight major climate change adaptation and mitigation on going and implemented in the Caribbean.

Resilience refers to "The capacity of social, economic, and environmental systems to cope with a hazardous event or trend or disturbance, responding or reorganizing in ways that maintain their essential function, identity, and structure, while also maintaining the capacity for adaptation, learning, and transformation" [1]. Resilience is seen as having an opposite relationship with vulnerability [10]. According to Hay [11], "Vulnerability and resilience are considered to be important integrating concepts when managing the local consequences of global changes". As a result, adaptation actions are geared towards reducing vulnerability and increasing resilience through improving climate-related knowledge and strengthening socio-economic systems and livelihoods [11–13].

Thought SIDS are not homogenous, they all have several specific characteristics that classify them as highly vulnerable. For example, their geographical location in hazard-prone regions, limited physical size, and a concentration of their population along coastal zones. According to the UN [14], these characteristics have propelled policymakers and scholars to highlight the need for special attention and support for SIDS in adapting to increasingly extreme weather events. In the case of the Caribbean, its geographic location in the Atlantic basin exposes the majority of the regional states to possible yearly hurricane impacts [15]. The UNWTO [16], cited that approximately 70% of the Caribbean population lives in coastal cities, towns and villages. Wilkinson, Twigg, and Few [6], expressed the Caribbean region's high level of public debts as a main vulnerability to development and increasing resilience.

Exposure depicts the extent and duration to which a system exists inside the scope of a hazard event and could be adversely affected [4,17]. Therefore, exposure depends on the system's location, geography, and climate change readiness [18]. Thus, for example, SIDS location is a critical factor to their climatic exposure [1,19]. According to Nurse et al. [1], SIDS' geography exposes them to crucial climate and ocean drivers that include tropical cyclones, rainfall, air and ocean temperature change, storm surge, and drought. Due to low elevations, many SIDS are increasingly susceptible to coastal erosion, flooding, and other extreme events [20]. Kelman [21] noted the importance to focus on the underlying development challenges instead of external threats which come up through discussing sensitivity and adaptive capacity.

Sensitivity is described as the degree to which a system is affected by climate change and is mainly dependent on intrinsic and contextual conditions [17]. A significant factor that gives rise to SIDS' sensitivity is the higher risk of economic instability [22]. SIDS limitations that contribute to their economic uncertainty are their insularity and physical size, their capacity to exploit new trading opportunities and increase the competitiveness of existing economic activities at a global scale [23]. SIDS higher risk of economic stability is due to their dependence on a narrow range of export products, limited opportunities for economic diversification, high transportation and communication costs, and high import content, particularly on food and fuel [24]. SIDS economies lack of economic diversification and are highly dependent on climate-sensitive sectors such as agriculture, fisheries, and tourism [1]. One event, for example, a tropical cyclone, can have a severe impact on the countries' Gross Domestic Product (GDP) and livelihoods [1]. Another factor cited in the literature that can increase sensitivity is a demographic shift where the majority of the population live in the most significant urban areas [25]. This trend can result in overcrowding and poor housing conditions, which leads to increase sensitivity to the impacts of extreme weather events [25].

The IPCC [4] defines adaptive capacity as "the ability of systems, institutions, humans and other organisms to adjust to potential damage, to take advantage of opportunities, or to respond to consequences". Betzold [19], concluded that the adaptive capacities of SIDS are limited by various factors such as lack of resources, institutional barriers, and public perception and awareness. Nunn [2], cited an increasing dependency on donor knowledge and funding as a hindrance to local communities' adaptive capacities. Significant institutional barriers include weak enforcement [26], poor coordination and communication at the national and local level [27], and lack of capacity [19].

Therefore, the abovementioned geographic and socio-economic factors cited from the literature characterizes SIDS' vulnerability and adaptive capacity to increase resilience. These features highlight SIDS' distinctiveness from other developing and developed countries and therefore require separate and context-specific vulnerability assessments [23,25]. In all international agreements, the United Nations [28] label SIDS as a "special" case for environment and development.

SIDS have been proactive in response to climate change challenges. Most notably, the donor community has reacted positively to SIDS adaptation challenges by contributing US$55.6 billion in ODA to Caribbean and Pacific SIDS between 1995 and 2015 [29]. Ac-

cording to Robinson [7], Allocation of funding to SIDS are highly skewed. Caribbean SIDS challenges related to extreme weather events are compounded by the lack of capacity and limited financing to effectively implement management strategies. Moreover, available resources continue to flow primarily to post-disaster activities instead of preemptive disaster risk reduction and improvement of coping capacity [30].

Since the mid 1990s, the Caribbean has experienced a significant increase in hurricane and tropical storm activities in the region. Barker [31], argued that an increase in Atlantic hurricane activity and the potential threat of intense tropical hurricanes in the future, poses a severe threat to the Caribbean region and its development. Forecast models show climate change would be responsible for a doubling of cyclone losses from US$26 billion per year to $53 billion by 2100 [32,33].

Climate change is arguably the most serious challenge to the development aspirations of the Caribbean region. The Emergency Events Database (EM-DAT) is a free and fully searchable database that contains worldwide data on the occurrence and impact of over 20,000 natural and technological disasters from 1900 to the present day. It records confirms the Caribbean has experienced 70 named tropical cyclones across 19 countries that devastated the region between the years 1980 to 2019. This paper analyses the impacts of tropical cyclones to the region between 1980 and 2019. It has three aims. First, it analyses the economic loss and damage sustained by the region. Second, it examines the most impacted sectors. Third, it obtains the perspectives of stakeholder in the Caribbean on the factors that hinders resilience.

## 2. Materials and Methods

Figure 2 shows the nineteen countries in the Caribbean that were selected to understand the individual and collective impacts on the region and respective countries and to highlight the challenges faced to combat against climate change. The map (Figure 2) also indicate that the nineteen countries were chosen using the Emergency Events Database EM-DAT database while five of those countries were selected based on availability of data to analyze the sectoral loss and damage from the Post Disaster Needs Assessments (PDNA).

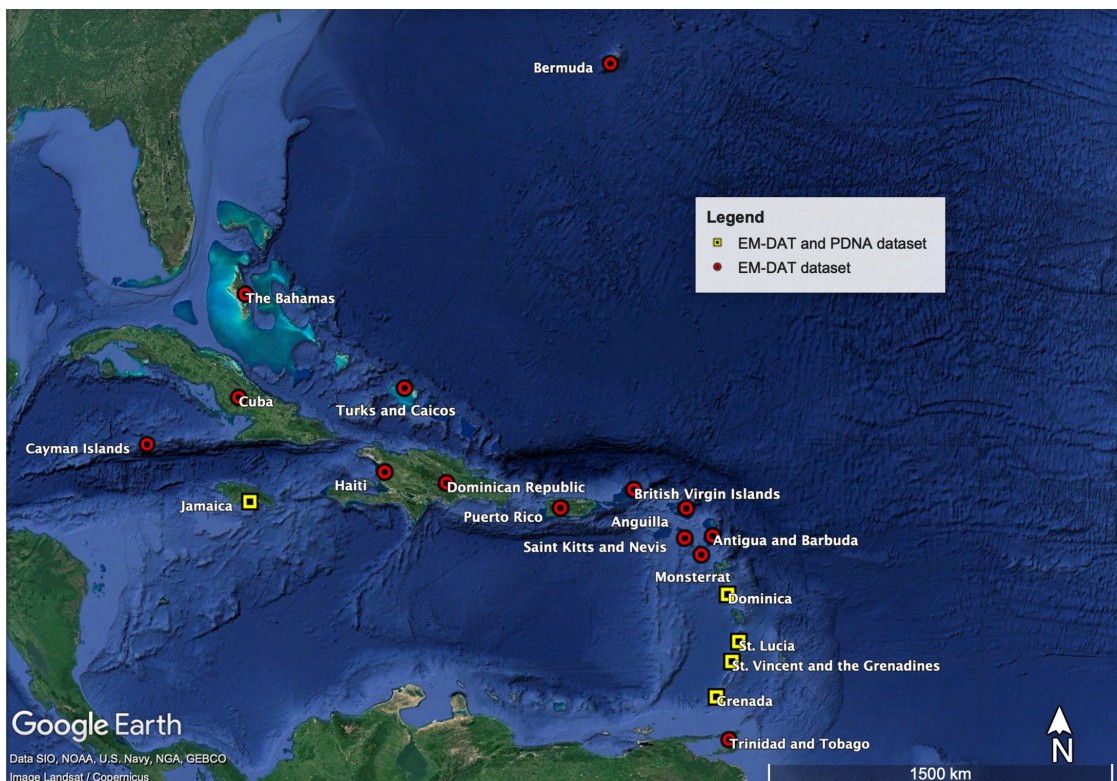

**Figure 2.** Map of Caribbean countries selected for the study.

Using a mix method approach, the main sources of data to inform this paper were from the Emergency Events Database (EM-DAT) and Post Disaster Needs Assessments (PDNA). The results were drawn from statistical analysis and content analysis.

For the quantitative approach, the required data to be analyzed was taken from the EM-DAT database and PDNA. The Emergency Events Database (EM-DAT) was launched in 1988 by the Centre for Research on the Epidemiology of Disasters (CRED). The main objective of the database is to serve the purposes of humanitarian action at national and international levels. EM-DAT contains essential core data on the occurrence and effects of over 22,000 mass disasters in the world from 1900 to the present day. The database is compiled from various sources, including UN agencies, non-governmental organizations, insurance companies, research institutes and press agencies. The steps taken to select the navigate the database and select the data are as follows:

- Firstly, the disaster list was selected from the database search options.
- Secondly, the period from 1980 to 2019 was inputted.
- Thirdly, the geographical region was selected as well as choosing the 19 individual countries.
- Fourthly, the disaster type (meteorological) and total damage and loss were selected. By clicking the search button, the data was generated.

The data was then analyzed using GraphPad prism (SD, CA, USA) software to categorize the data into graphs.

Desk-based research was conducted to collect PDNA documents for St. Lucia, St. Vincent and the Grenadines, Jamaica, Dominica, and Grenada corresponding to the period from 2000 to 2019. These countries and time period were chosen based on the availability of the data. The economic loss and damage data from the productive, infrastructure, and social sectors were collected. Two sectors each were chosen from the productive, the infrastructure, and the social sectors. These include tourism and agriculture (productive sector), transport and utility (infrastructure sector), and housing and health (social sector). For this paper, the term utility sector is used in reference to the telecommunication, electricity, and water sectors. The complete dataset was then divided into two periods, 2000–2009 and 2010–2019 for comparative analysis purposes. GraphPad prism (SD, CA, USA) was used to analyze the data while descriptive statistics generated graphs to facilitate the explanation of the findings. The complete dataset from the EM-DAT were analyzed to ascertain the decadal economic loss and damage in the region, and the category of tropical storms between 1980 and 2019. The dataset from the PDNA were analyzed to understand and compare the most affected sectors from 2000 to 2009 and 2010 to 2019. The complete datasets were converted to 2019 constant U.S Dollars. The mean and standard error of mean (SEM) were calculated to show the variability within the sampled dataset.

The qualitative approached included eighteen semi-structured interviews which were conducted at two different time periods. Seven interviews were conducted in St. Vincent and the Grenadines (SVG) in August 2018 while eleven were conducted at the Conference of the Parties Twenty-Four (COP24) in December 2018. The interviewees included public and private stakeholders in St. Vincent and the Grenadines integrally involved in climate change and development issues and climate change experts from Antigua and Barbuda, The Bahamas, Barbados, Belize, Jamaica, St. Lucia, along with representatives from Caribbean Community (CARICOM), CCCCC, OECS, and Climate Analytics working on climate change related issues. Content analysis was used to examine the data collected from the interviews on the challenges and vulnerabilities faced by the region to build resilience to TC and climate change holistically. The data ascertained from the interviews were also used to design a matrix to illustrate the potential damage incurred by extreme weather events on the different sectors.

The selected methods have two primary limitations. Using the EM-DAT and PDNA as the main sources of data are subject to under and/or over-reporting which can affect the outcome of the results. Case-study qualitative research provides an opportunity to produce rich narrative intertwined with theory to answer critical questions [34]. However, this approach has its limitations. Case study research, particularly small N studies, as in

the case of this study, suffers from two primary limitations: indeterminacy and selection bias [35,36].

## 3. Results

### 3.1. Quantitative Analysis

Six-seven (67) TC were recorded from the data collected between 1980 to 2019 (Table 1). The most frequent TC occurring in the region are tropical storms accounting for 34% while category 5 hurricanes are the least with 4%. Figure 3 shows that over the 40 years, the total loss and damage to the region is approximately US$145 billion (2019 constant U.S dollars). The graph (Figure 3) indicates how the economic loss and damage were broken down into the different tropical storms and the varying categories of hurricanes that impacted the Caribbean region between the period of 1980 to 2019. As seen in Figure 3 tropical storms and categories 1, 3, and 4 hurricanes occurred during between 1980–2009 while category 5 hurricanes were seen in the region between 2010–2019. The economic loss and damage sustained from the tropical storms and categories 1–4 hurricanes were approximately US$45 billion. The tropical storms and categories 1–4 hurricanes accounted for 96% of all TC included in the study. On the other hand, the economic loss and damage from the category 5 hurricane amounted to approximately US$100 billion. Although the category 5 hurricanes accounted for 4% of the total TC in the region between 1980–2019, the impact has cost the region substantially more than all the other types of TC included in the study. The least economic loss and damage was sustained from category 2 hurricanes of approximately US$3.1 billion while the most impacts were sustained from category 5 hurricanes costing approximately US$100 billion. Tropical storms are the most frequently occurring TC in the region incurring approximately US$6.2 billion between 1980 to 2019. The more intense hurricanes which include categories 3–5 was less frequent with an economic loss and damage to the region of approximately US$129 billion.

**Table 1.** Types and number of TC occurred in the region between 1980–2019.

| Type of TC | Number of TC |
| --- | --- |
| Tropical Storm | 23 |
| Category 1 | 10 |
| Category 2 | 8 |
| Category 3 | 11 |
| Category 4 | 12 |
| Category 5 | 3 |

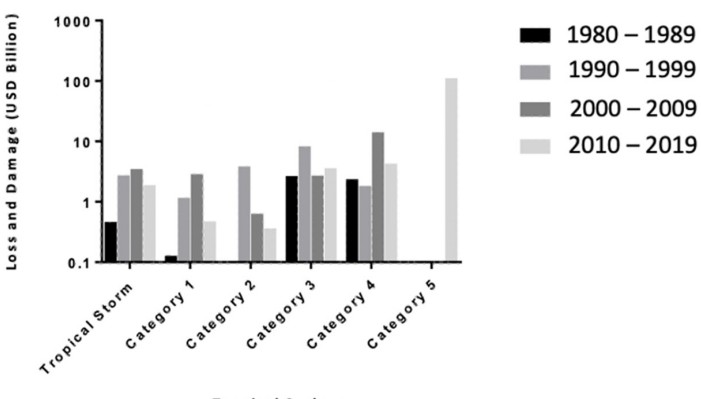

**Figure 3.** Economic loss and damage incurred by the different categories of TC selected in the Caribbean between 1980–2019.

Figure 4 shows the Mean ± SEM of the economic loss and damage sustained by the region between 1980 to 2019. The mean values are approximately US$261 million,

US$490 million, US$493 million, and US$3.6 billion for the periods 1980 to 1989, 1990–1999, 2000–2009, and 2010 to 2019 respectively. The short Error bars seen in Figure 4 shows a higher accuracy of the plotted average of the economic loss and damage illustrated in the study. The period between 1980 to 1989 illustrates the lowest mean of approximately US$261 million while the period between 2010 to 2019 shows the highest mean of approximately US$3.6 billion. The highest mean corresponds with the economic loss and damage caused by category 5 hurricanes as seen in Figure 3 for between 2010 to 2019.

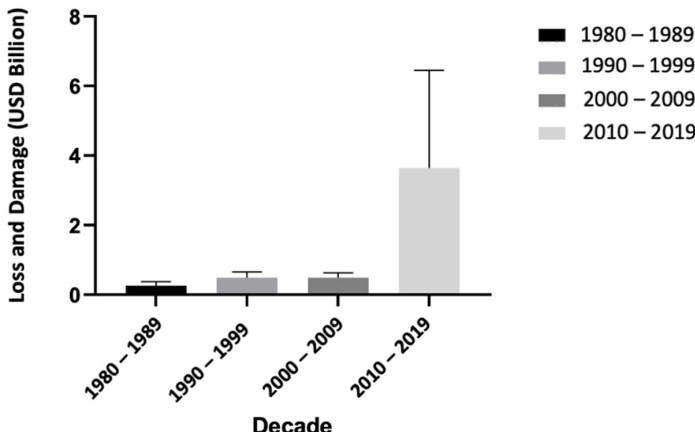

**Figure 4.** The mean and SEM of the economic loss and damage sustained in the Caribbean by TC from 1980–2019.

Figure 5 compares the economic loss and damage incurred between 2000 and 2009 and 2010 and 2019 by illustrating the sectors that are most affected by the impacts of extreme weather events and expressed as a percentage. The sectors most affected are the agriculture, housing, transport and utility. Between 2000 and 2009 the housing sector sustained 39.9% economic impacts from extreme weather events, followed by the utility sector (19.1%), and the agriculture sector (16.3%). Between 2010 and 2019 the transport sector (39.9%) sustained the most impacts, followed by the housing sector (25.1%), and the agriculture sector (16%). The sector least affected in both decades was the health sector with 1.21% (2000–2009) and 1.12% (2010–2019) respectively. The figure shows that while there was a significant reduction in the economic impacts to the housing (14.9% reduction) and utility (5.9%) sectors, there was a substantial increase in the economic impacts to the transport sector (22.4% increase) between 2010 and 2019 as compared with 2000 and 2009.

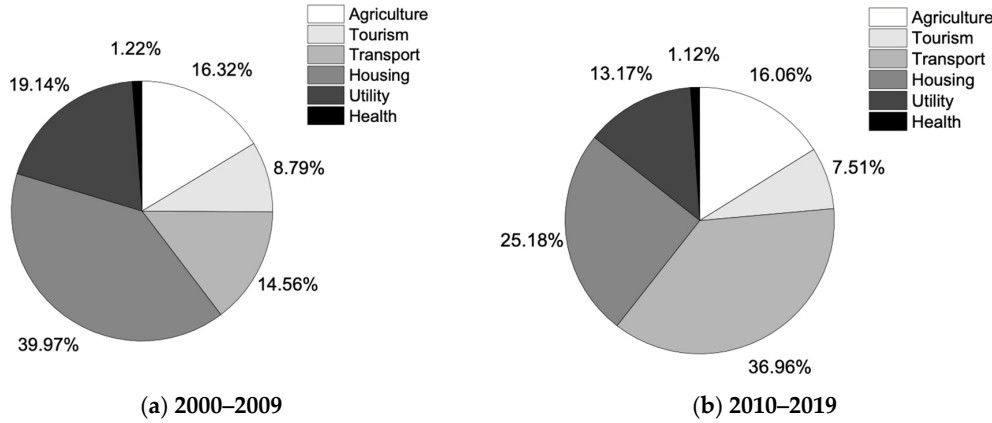

(**a**) 2000–2009        (**b**) 2010–2019

**Figure 5.** Loss and damage incurred from the selected Caribbean Countries (St. Vincent and the Grenadines, Grenada, Jamaica, Saint Lucia, and Dominica): (**a**) percentage of loss and damage on the agriculture, tourism, transport, housing, utility, and health sectors from 2000–2009; (**b**) percentage of loss and damage on the agriculture, tourism, transport, housing, utility, and health sectors from 2010–2019.

### 3.2. Qualitative Analysis

Table 2 shows eleven potential damage and cascading effects caused by TC and how they interrelate with the different sectors. Analyzing the agriculture sector, the correlation shows that the impacts of tropical cyclones can have major implications to the country's food security, especially in the case of the Caribbean SIDS where agriculture is one the main pillars of their economy. Therefore, with the instability of food security, this can have a spill off effect on other sectors such as tourism, transport, health, and housing. These negative spill offs will contribute to more poverty, higher importation food bill, general decrease in health, and lack of tourist visiting the islands due to higher commodity cost. The table considers twelve potential damages and cascading effects and their adverse impacts on the six sectors.

**Table 2.** Link between the sectors and the damages and cascading effects caused by TC.

| | | Sectors Impacted by Tropical Cyclones | | | | | |
| --- | --- | --- | --- | --- | --- | --- | --- |
| | | **Agriculture** | **Tourism** | **Transport** | **Utility** | **Health** | **Housing** |
| **Potential damages and cascading effects** | Increase in food security issues | ✓ | ✓ | ✓ | | ✓ | ✓ |
| | Loss of livelihoods | ✓ | ✓ | ✓ | | | ✓ |
| | Damage to transport facilities | ✓ | ✓ | ✓ | ✓ | ✓ | ✓ |
| | Water shortages | ✓ | ✓ | | | ✓ | ✓ |
| | Increase water & mosquito borne diseases | ✓ | ✓ | | | ✓ | ✓ |
| | Electricity & Telecommunication shortages | | ✓ | | ✓ | ✓ | ✓ |
| | Loss in commercial agriculture | ✓ | ✓ | ✓ | | | ✓ |
| | Decrease in foreign investments and revenues | ✓ | ✓ | ✓ | ✓ | | ✓ |
| | Damage to health facilities | | ✓ | ✓ | | ✓ | ✓ |
| | Increase cost of living | ✓ | ✓ | ✓ | | | ✓ |
| | Increase cost to rebuild and restore transport facilities | | ✓ | ✓ | | | |

Figure 6 gives a visual illustration of what were the main concerns from the interviews. The majority of the interviewees highlighted high public debt, adaptation cost, and the increasing loss and damage from more intense TC as reasons why building resilience is extremely challenging for the region. One interviewed mentioned "we are battling several disasters on multiple fronts. We are battling high levels of debt, erosion of preferences, lack of access to concessional finance". The interviewee further expressed that "it's a lot for small economies to deal with and again every time we are impacted by a hydro-meteorological disaster it sets the development of the country back by decades". Another interviewee from St. Vincent and the Grenadines stated "Increasing public debts and so much national development needed in country, the government main focus is what can be done within the five-year election cycle". Another interviewee lamented that "adaptation cost to systematically build resilience is beyond the capacity of Caribbean SIDS". An interviewee from St. Lucia mentioned "the increased intensity of hurricanes has the ability to erase decades of development from a single category 5 hurricane". The interviewee further used Dominica as an example as the loss and damage caused by the impacts of hurricane Maria.

When asked about what measures are most important to build resilience, the majority of the respondents suggested the need for increased access to adaptation financing, increased in Early Warning Systems (EWS) and a revision of SIDS income status classification which is a major handicap to the region. One interviewee from Jamaica stated that "increased access to adaptation financing is necessary for Caribbean SIDS to have a chance in the face of climate change". The interviewee further stated that "we can see the impacts of climate change through the increased intensity of hurricanes and sea-level rise". Another interviewee added to the fact that "countries are getting international funding for climate change but some projects fall apart because funds run out or because of silos in

the government then project funding isn't does as much as it can". One interviewee from The Bahamas stated that "climate financing is indispensable" and further noted that "in order to access those financial flows you have to have the systems in place to do all the things you need to apply for it, to use it, to be able to report on it, and be in a position on an on-going basis to constantly update and upgrade the amount of money that you need base on the circumstances as they change. Another interviewee highlighted the need to prioritize EWS by stating "the loss and damage from hurricanes are devastating and we need to close the gap by introducing more technology including EWS to given ourselves a chance". The interviewee further stated that "too often we have to start from scratch when a hurricane hits the region".

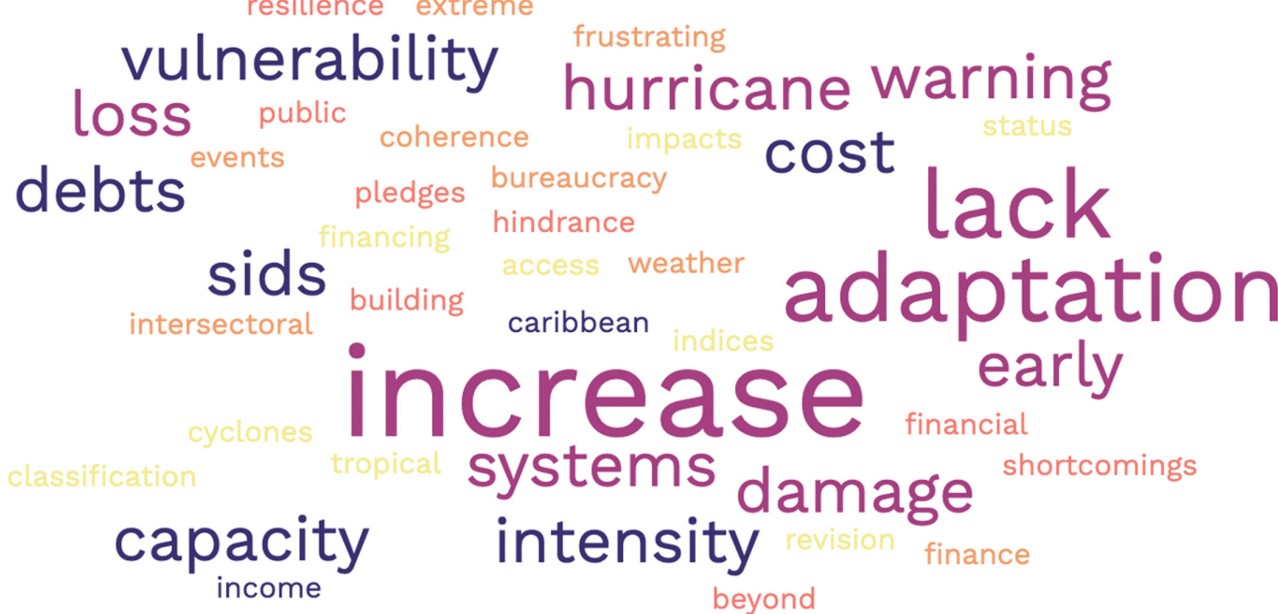

**Figure 6.** Keyword analysis map from interview.

When asked about how Caribbean countries can better prepare to build resilience, the majority of respondents mentioned intersectoral coherence, targeted capacity building, and increased awareness. one of the interviewees from Antigua and Barbuda mentioned that many sectors are devastated when there is a hurricane. The interviewee further states that "it's about time government ministries recognize the importance of planning together and move away from the silo approach". Another interviewee from Belize brough to attention that "within the region, we are focused on building climate resiliency but the coordinating effort isn't really there and that's one of the hurdles we need to get over in the region". In terms of targeted capacity building, one interviewee stated that "you have to first build your capacity in order to identify what you need. Without taking that first step of building national capacity be it for adaptation or mitigation, all your efforts will be in vain". Another interviewee underscored the importance of capacity building by stating "it is essential to access financing" and without the capacity needed to access financing "you don't have a holistic approach pertaining to climate change".

To have a sense of appreciation of the challenges faced by the Caribbean region, the economic loss and damage sustained and the keywords from the word map are combined into 4 major topics. (1) The inherent disadvantaged of Caribbean SIDS income status classification and high public debt. (2) The need for increased access to adaptation financing for SIDS. (3) The need for increased EWS.

## 4. Discussion

### 4.1. The Inherent Disadvantage of Caribbean SIDS Income Status Classification and High Public Debts

The analysis of the interviews revealed stakeholders' dissatisfaction of the classification of the income status of Caribbean SIDS. The majority of the interviewees lamented that the "false" classification of Caribbean SIDS, coupled with their high public debts hinders the regions effort in reducing vulnerability to the impacts of climate change. The World Bank has historically classified every economy as low, middle or high income. It further specifies its classes of countries too, into low, lower-middle, upper-middle and high-income economies. The World Bank uses Gross National Income (GNI) per capita as the basis for this classification because it views GNI as a broad measure that is considered to be the single best indicator of economic capacity and progress. Middle-income countries (MICs) are nations with a per capita GNI between \$1006 and \$12,235 [37]. By using GNI per capita to classify income status, the majority of Caribbean SIDS falls in the category of lower-middle income economies (per capita GNIs between \$1006 and \$3955) and upper-middle income economies (per capita GNIs between \$3956 and \$12,235) [37]. For Caribbean SIDS to strengthen their resilience by reducing their vulnerabilities, adaptation financing is necessary. One interviewee mentioned that adaptation programs and policies are driven by financing. While another cited that there are countries in the region that cannot access some funding sources because of their economic classification, while others countries accessing loans for climate change adaptation are already being choked with high public debts. The interviewees reflected on the results of Figures 3 and 4 and zoomed into the years between 2010 to 2019. Interviewees expressed the increased frequency and intensioty of tropical cyclones between the period of 2010 to 2019 and made mention of Hurricanes Irma and Maria as two of the most devastating in the region.

The income status classifications pose a huge challenge to gaining financial resources for SIDS in response to climate change. Seventy percent of the interviewees acknowledged the classification of the majority of the countries are "false" because of the countries' small size, vulnerabilities to external shocks, and their high dependence on climate-sensitive sectors like agriculture and tourism. The 2017 hurricane season exposed the deficiencies of the international aid and assistance mechanism. Hurricanes Irma and Maria, both category five storms caused substantial destruction in the regions. The income status classifications meant that the countries most impacted were ineligible to access aid or concessional loans from bilateral and multilateral financing providers [38]. For example, Antigua and Barbuda suffered losses and damages in excess of US\$222 million [39], yet are limited to access of non-concessional emergency finance from the International Monetary Fund (IMF) in response to disasters [38]. Another major setback for Caribbean SIDS with the label of high-and-middle-income status is that they will soon be deemed ineligible for Official Development Assistance (ODA) [38]. Interviewees focused on Figure 4 to highlight the dire state of the region to the various sectors with the agriculture, transport, and housing sectors heavily impacted. One interviewee lamented the financial limitations faced in the region, stating the need to revive the economy after the impact of a hurricane. The interviewee further stated that with the extensive loss in the agriculture sector and the major damages to the transport sector it basically hinders the livelihoods of many persons and directly has a negative impact on the tourism sector.

Another major disadvantage to Caribbean SIDS in increasing resilience to the impacts of climate change is high public debt. The International Monetary Fund [40], reported that most of the Caribbean SIDS are highly indebted. The implications of Caribbean SIDS in such a peculiar economic position will have significant adverse effects on their development, which compliments increasing resilience and reducing vulnerability. In this regard, authors such as Bourne [41] argue for changes to the eligibility criteria for SIDS to access concessional development finance, and adaptation finance. Bourne [41] suggested that the current criteria are a disadvantage for SIDS in their access to such critical financing.

### 4.2. Need for Increased Access to Adaptation Financing for SIDS

The interviews have revealed the stakeholders' dissatisfaction with the current levels of international adaptation financing and their experience accessing it. Seventy-five percent of the interviewees acknowledged that accessing international financing is a challenge for Caribbean SIDS. Sixty percent cited their frustration with the funding mechanism systems for which the volume of paperwork required for funding application and the administration fees taken by implementing agencies as issues of major concern. The view-point from the interviewees is that such issues strain the already limited human resources in SIDS and lessen the monies available for actual adaptation. Several interviewees highlighted that it is easier to access bilateral financing than multilateral financing.

The Caribbean has seen an increased in extreme weather events over the past decade. Between 2017 and 2019 there has been three category five hurricanes that devastated the region. Hurricane Irma and Maria in 2017 caused significant economic loss and damage to Dominica and Antigua and Barbuda. Dominica reported an overall economic loss and damage of US$1.3 billion or 224% of their GDP sustained from the impacts of hurricane Maria [42]. The country sustained major economic loss and damage in the housing (38%), tourism (19%), and agriculture (17%) as a percentage of their GDP [42]. Antigua and Barbuda reported an overall economic loss and damage of US$222 million or 9% of their GDP, with 95% of the housing sector damaged in Barbuda [40]. Hurricane Dorian (2019) caused considerable damage to the Bahamas. The island reported an estimated US$3.4 billion in damages and other impact [43].

One interviewee cited that "the cost of adaptation is high, but the cost of inaction is higher" and that countries are dependent on adaptation financing to reduce their vulnerabilities to the impacts of climate change. The statement was made in reference to the intense extreme weather events experienced in recent years in the Caribbean. SIDS, especially through the Alliance of Small Island States, an alliance solidifying the voices of SIDS inside the United Nations framework, have called for "scaled-up, new, additional, and predictable financial resources" so as to adequately adapt to climate change with the current levels of international adaptation financing are insufficient [44]. The World Bank [45] projected that the costs for adapting to a 2 °C warmer world between 2010 and 2050 could range from US$70 billion to more than US$100 billion per annum. However, the climate finances committed by developed to developing countries in 2014 totaled to approximately US$25 billion [46], roughly one-quarter of what is suggested to be required annually.

Though Caribbean SIDS have expressed dissatisfaction with the bureaucracies involved with accessing adaptation financing, Caribbean SIDS have been positioning themselves to access financing directly. Through the Green Climate Fund (GCF), countries are putting steps in place to qualify as implementing agencies.

### 4.3. Need for Increased EWS

The majority of interviewees mentioned the need for increased ESW. In the Caribbean, the North Atlantic hurricane season brings an average of 10 named storms per year, with an average of six making it to hurricane strength, and 2.5 becoming major hurricanes (Category 3 or higher) [47].

In the case of hurricane Maria its sustained maximum winds were 90 mph as of 5 am Atlantic time on Monday 18th September 2017, putting it at the high end of a Category 1 hurricane. However, in approximately 15 h, winds rose to 160 mph, almost 50 mph higher than forecast the previous night, accelerating Maria to a Category 5 and catching many residents off guard [47,48] (NHC, 2019; NOAA, 2017). The Sendai Framework for Disaster Risk Reduction 2015–2030 issued the global challenge to substantially increase access to Multi-Hazard Early Warning Systems (MHEWS) as well as disaster risk information and assessments by 2030. To strengthen EWS in the region, one project in 2018 focused on Strengthening Hydro-Meteorological and Early Warning Services in which the key deliverables were development of regional strategy to strengthen and streamline early warning and hydromet services and institutional strengthening and streamlining of early

warning and hydromet services [49]. Such initiatives by organizations, in this case Climate Risk and Early Warning Systems (CREW) contribute to charting the way forward to building resilience in the region.

Another major initiative that the Caribbean region can capitalize on is the new United Nations target which states "In 5 years, everyone on Earth must be protected by early warning for extreme weather, climate change." [50]. The UN Secretary-General stated "We must invest equally in adaptation and resilience. That includes the information that allows us to anticipate storms, heatwaves, floods and droughts". He further states that "Early warnings and action saves lives" and It's unacceptable that a third of the world's people, mainly in least developed countries and SIDS are not covered by early warning systems.

The results show the impact of TC and the substantial loss and damage to the countries' productive, social, and infrastructure sectors. Therefore, taking into consideration the vulnerabilities of the Caribbean SIDS, special focus should be placed on the region for EWS. One interviewee stated what they refer to as the three "U" of climate change; unfamiliar, unprecedented, and urgent. The interviewee further stated that TC are now following an unfamiliar pattern, that their intensity is unprecedented, and the response is urgent. The Prime Minister of St. Vincent and the Grenadines in his address at the 73rd United Nations General Assembly in 2018 stated the need for small state exceptionalism to be placed at the center of the global discourse and responses addressing climate change [51]. The Prime Minister of St. Vincent and the Grenadines [51] further stated that "major emitters that fail to set and honor ambitious mitigation pledges are committing a direct act of hostility against the SIDS and we ought to resist the recklessness of those major emitters who are acting against our interest".

Also, as shown from the interviews, the need for a comprehensive EWS must include lessons learned from past events, in order to continually improve responses ahead of future weather, climate, water and environmental related hazards. They stated that such an EWS should also include agreed response plans for government, communities and people, to minimize anticipated impacts.

## 5. Conclusions

Climate change can derail the socio-economic and development strides that the Caribbean has made over the past decade. Climate change adds "considerable stress" to individuals, social groups, communities, sectors, countries and regions. Its progression cannot be halted in the short-term, hence the region needs to adapt to it adverse effects. Adaptation to climate change requires substantial monetary investment [52]. Caribbean SIDS have to embrace a more profound 'regionalism' to reduce their vulnerability to economic and environmental shocks [53]. One strategy that the Caribbean has implemented to adapt to extreme weather events is to improve EWS for TC. This is a 'no or low-regrets' option for adaptation to climate change. It provides benefits under any range of climate change scenarios and cope with high levels of uncertainties over future climate change directions and impacts [54]. Investments in early warning services should be integrated with current disaster risk reduction and other development strategies (i.e., poverty reduction and education) to improve the effectiveness of national systems for managing disaster risk [55,56].

Although the capacities for Caribbean SIDS to manage disaster risk are improving, they lack the strong institutions and systems needed to anticipate and cope with hazards, whereas, vulnerability levels are high because of limited employment opportunities, inequality, and difficult market conditions [57,58]. The Caribbean islands face several limiting factors that include their vulnerability and exposure of the islands to natural hazards on a yearly basis, their high debt, and their lack of resources which makes it nearly impossible to obtain sustainable development. To strengthen the position of the Caribbean region in building resilience to extreme weather events, the following recommendations should be considered: (1) Revise the per capita national income as an eligibility criterion for access to concessional development finance assistance. As an indicator, it does not adequately reflect

the Caribbean SIDS deficiencies. (2) A comprehensive EWS for the countries in the region and (3) Consideration of debt relief for countries who have suffered major loss of physical and economic infrastructure through extreme weather events.

**Funding:** This research received no external funding.

**Data Availability Statement:** Not applicable.

**Acknowledgments:** I would like to thank all the interviewees for their invaluable contribution to the paper.

**Conflicts of Interest:** The author declares that there is no conflict of interest.

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
