# Peer review of "Climate Change and the Caribbean: Challenges and Vulnerabilities in Building Resilience to Tropical Cyclones"

_climate, doi:10.3390/cli10110178_

Round 1
Reviewer 1 Report
General comments
This study focuses on the climate change vulnerability in the Caribbean. The topic is very interesting and it has practical importance, particularly for disaster-prone Caribbean areas. Though this topic is quite interesting and of practical significance, the manuscript suffers from a very odd presentation style, poor introduction, poor methodology, weak results, and discussion section. The author claims that this study also focuses on hindrances of resilience, but it is actually vulnerability, not resilience. The author did a lot of work but due to poor presentation, it fails to convey the exact message of the research. It looks like an assignment paper, not a research paper.
Title
The title should be replaced by “Assessment of vulnerability in the face of tropical cyclones: Evidence from Caribbean region”. Because this manuscript is dealt with vulnerability, not resilience. So, using “resilience” will make confuse readers.
Abstract
It is not well structured. The essential points are missing. There is little connection between the body text and the abstract. The author is advised to revise the abstract by following IMRAD style by avoiding unnecessary text. The reader wants to see what is the exact findings of the study, rather than the background.
Introduction
-The introduction is poor.
-Though, the author introduces the issue, its importance, and its practical implication, there are huge unnecessary texts. The author is advised to remove unnecessary texts.
-The author could not focus on the research gap, research questions, and hypothesis. So, it is a question “why is this study necessary? Is it really a research item?
-There is no coherence in the description.
-Though, the author explained the major components of vulnerability (exposure, sensitivity, and adaptive capacity). But, there is a huge inconsistency in the text under this section.
-Overall, this section fails to meet criteria of an introduction.
Materials and Methods
-This section is also unstructured and poor.
-The author should explain the steps of the research properly, so that anybody can replicate it.
-This section should be divided into several sections, like geographical features, sampling methods and sample size determination, data sources, measurement of vulnerability, approaches for measuring economic loss, determinants or influencing factors by regression model, and perception of people on the causes of vulnerability.
-Unfortunately, the sample size is 18 that does not meet the criteria of a standard sample size. Besides, no sampling determination method is used for it.
-The whole methodology is faulty. So, the author is advised to conduct the study again by following proper research steps with an accepted sample size.
-The researcher community will not accept this descriptive statistics based vulnerability measurement. There are many methods for measuring vulnerability. You can follow any of them.
Results
-The results will not be accepted due to faulty methodology.
-The author is advised to present the results in several sub-sections by following a revised methodology.
Discussion
-How can we find out the discussion without proper methodology?
-This section is also very poor and haphazard.
-There is no planning for presenting the findings properly.
-There is little connection between the research objective and the findings.
-The author is advised to revise the whole section by following a new methodology.
Conclusion
This section is not structured. It should be revised by adding key findings, recommendations, and practical implications.
References
Need to check the whole section and follow the journal style.
Author Response
Dear Editor,
Please see refusal for reviewer 1

Reviewer 2 Report
Please see the attached document.

Author Response
Dear reviewer 2, please see attached article.

Reviewer 3 Report
Dear Editor,
Please find my review of a study " Climate change and the Caribbean: Challenges and vulnerabilities in building resilience to tropical cyclones" by Lewis Todd Clint submitted to Climate for consideration for possible publication.
This study examines the impacts of tropical cyclones on the Caribbean region between 1980 to 2019 and it aims to analyse the economic loss and damage sustained by the region, identify the sectors most impacted, and ascertain the perspectives of key stakeholders on the factors that hinder resilience. It was found that the economic loss and damage caused by tropical cyclones in the region has been gradually increasing between 1980 to 2009, followed by a drastic increase between 2010 to 2019. It was also found that finds the most impacted sectors are the agriculture, housing, transport and utility.
The subject of this report is suitable for Climate and it could be published after suggested revision.
Dear Author,
This is a valuable study which addresses urgent needs to improve resilience of SIDS to tropical cyclones. Please find below my comments for your consideration.
Abstract
Please consider some editorial changes to improve presentation.
You may wish to introduce an acronym TC for a tropical cyclone, as you use it often in the manuscript.
Line 11: impacts of tropical cyclones, a rapid onset event. => impacts of tropical cyclones, rapid onset events OR impacts of tropical cyclones, rapid onset hazards.
Lines 11 -14. This statement is too long; consider splitting in into two sentences. Also , "This paper analyses the impacts of tropical cyclones on the region between 1980 to 2019 and it aims to analyze …" First, you use "analyse" twice and it does not read well; second – use UK analyse or US analyze, but not both.
Lines 18 – 19: "It finds the sectors most impacted are the agriculture, housing, transport and utility sectors. " As above, you use "sectors" twice, please revise to improve it.
1. Introduction
It would be beneficial to update references to more recent publications. For example, you cite IPCC reports from 2014:
IPCC. (2014). Annex II: Glossary, In Climate Change 2014: Impacts, Adaptation, and Vulnerability. Part B: Regional Aspects. Contribution of Working Group II to the Fifth Assessment Report of the Intergovernmental Panel on Climate Change (Eds, Barros, V. R., Field, C. B., Dokken, D. J., Mastrandrea, M. D., Mach, K. J., Bilir, T. E., Chatterjee, M., Ebi, K. L., Estrada, Y. O., Genova, R. C., Girma, B., Kissel, E. S., Levy, A. N., MacCracken, S., Mastrandrea, P. R. and White, L. L.) Cambridge University, Press, Cambridge and New York, pp. 1757-1776.
Nurse, L.A., McLean, R.F., Agard, J., Briguglio, L.P., Duvat-Magnan, V., Pelesikoti, N., Tompkins, E., & Webb, A. (2014). Smallislands. In V.R, Barros., C.B, Field, D.J, Dokken, M.D, Mastrandrea, K.J, Mach, T.E, Bilir, M, Chatterjee, K.L, Ebi, Y.O, Estrada, R.C, Genova, B, Girma, E.S, Kissel, A.N, Levy, S, MacCracken, P.R, Mastrandrea, & L.L, White (Eds.). Climate Change 2014: Impacts, Adaptation, and Vulnerability. Part B: Regional Aspects (pp. 1613-1654).
I encourage you to read IPCC 2022 and update your manuscript with more recent information.
Line 41: For example, rapid onset events such as hurricanes => For example, rapid onset events such as tropical cyclones (also known as hurricanes in the Atlantic basin)
Line 64: (Source: Author). If this figure was produced by you, why do you need to include the source? Please check Authors Guidelines.
Lines 126 – 133: Suggest also mentioning CREWS https://www.crews-initiative.org/en as it assists the Caribbean SIDS with improving early warning systems for tropical cyclones.
2. Material and Methods
Line 183: were calculated for figure 2.
=> Figure 2. Capitalise word Figure here and throughout the text.
3. Results
Line 220: by varying tropical storm anf the different categories of hurricanes ranging from category => by varying tropical storm and the different categories of hurricanes ranging from category
Line 232: A closer look at the The figure also highlights the The figure also shows that least … Please correct
4. Discussion.
Line 347: the statement was made in reference to the => The …
Suggest strengthening this section adding discussion about an ambitious new UN target : everyone on Earth protected by “early warning systems against increasingly extreme weather and climate change” within five years. https://www.climatecentre.org/8111/un-within-5-years-everyone-on-earth-must-be-protected-by-early-warningfor-increasingly-extreme-weather-and-climate-change/
This new UN target is directly applicable to reducing vulnerability and increasing resilience of SIDS, including to tropical cyclones.
While importance of early warning services is highlighted in Conclusions, lines 396 - 403, these statements are not supported by Discussion. Please add a relevant section.
This reviewer recommends accepting the manuscript for publication after a minor revision.
Yours faithfully,
The Reviewer
Author Response
Dear reviewer 3,
Please see attached document.

Round 2
Reviewer 1 Report
The author did not address any comments.
Reviewer 2 Report
Thank you for the completed revisions. I believe the manuscript is now suitable to publish.